# Social Stimulation by the Owner Increases Dogs’ (*Canis familiaris*) Social Susceptibility in a Food Choice Task—The Possible Effect of Endogenous Oxytocin Release

**DOI:** 10.3390/ani12030296

**Published:** 2022-01-25

**Authors:** Anna Kis, Henrietta Bolló, Anna Gergely, József Topál

**Affiliations:** Research Centre for Natural Sciences, Institute of Cognitive Neuroscience and Psychology, Magyar Tudósok krt. 2, 1117 Budapest, Hungary; bollo.henrietta@ttk.hu (H.B.); gergely.anna@ttk.hu (A.G.); topal.jozsef@ttk.hu (J.T.)

**Keywords:** *Canis familiaris*, dog, human influence, oxytocin, social priming

## Abstract

**Simple Summary:**

Intranasal administration of oxytocin has been proposed to be an effective way of improving several socio-cognitive skills in humans. There is evidence that dogs show human-analogue social behaviours and socio-cognitive capacities. Furthermore, recent studies have demonstrated that the oxytocin system is related to human-directed social behaviour in dogs. Some human studies suggest that pre-treatment with social stimuli (eye contact, touch) has similar behavioural effects because these cues stimulate oxytocin release. In the present study, we show that both social stimulation by the owner as well as intranasal oxytocin administration increases dogs’ social susceptibility in a food choice task. This means that dogs after both treatments (as compared to control conditions) were more prone to conforming to a human demonstrator’s counterproductive choice (smaller amount of food), giving up their natural preference. These results highlight important mechanisms of social tuning between dogs and humans.

**Abstract:**

Recent evidence suggests a human-like susceptibility to social influence in dogs. For example, dogs tend to ignore their ‘natural’ preference for the larger amount of food after having seen a human’s explicit preference for a smaller quantity. However, it is still unclear whether this tendency to conform to the partner’s behaviour can be influenced by social stimuli and/or the neurohormone oxytocin as primers to prosocial predispositions. In Experiment I, eighty two dogs were tested using Prato-Previde et al.’s food quantity preference task. In Experiment I, we investigated in a 2 × 2 design how (i) a 10-minute-long social stimulation by the owner versus a socially ignoring pre-treatment as well as (ii) on-line ostensive communications versus no communication during task demonstration affect dogs’ (N = 82) choices in the abovementioned food choice task. Results indicate that the owners’ pre-treatment with social stimuli (eye contact, petting) increased dogs’ susceptibility to the experimenter’s food preference, but the salient ostensive addressing signals accompanying human demonstration masked this social priming effect. In Experiment II, N = 32 dogs from the subjects of Experiment I were retested after oxytocin (OT) or placebo (PL) pre-treatments. This experiment aimed to study whether intranasal administration of oxytocin as compared to placebo treatment would similarly increase dogs’ tendency to re-enact the human demonstrator’s counterproductive choice in the same task. Results showed an increased susceptibility to the human preference in the OT group, suggesting that both socially stimulating pre-treatment and the intranasal administration of oxytocin have similar priming effects on dogs’ social susceptibility.

## 1. Introduction

One of the key characteristics of human group living is social conformity—the act of matching attitudes, beliefs, and behaviours to group norms in general and to another person’s attitudes in particular [1]. Importantly, the act of matching often takes place spontaneously, without conscious control over the interactants’ actions [2,3]. Although many assume that social susceptibility processes are based on uniquely human skills, non-human animals have also been reported to adjust their behaviour to previously observed social actions of their interactant [4]. This so-called social priming effect has, in turn, been shown in humans to influence several social behaviours including affiliation [5] and cooperation [6].

From those non-human species that have so far been used to study the analogues or homologues of human social behaviour, the domestic dog is especially important, as during domestication, it has adapted to the cognitively and socially challenging human environment [7,8]. Family dogs show several functionally human-like social skills and appear to be human-tuned in their social behaviour skills [9,10] and this enables them to achieve a higher level of synchronization when interacting with humans [11]. Dogs are equipped with skills necessary to establish behaviour synchrony [12,13], and they can efficiently use human behaviour as a cue for performing functionally equivalent ‘imitative’ response [14,15]. There is also empirical evidence suggesting that they can use the emotional information provided by a human about a novel object to guide their own behaviour towards it [16,17]. Furthermore, dogs are willing to conform to a human’s (both their owner’s and an unfamiliar experimenter’s) explicit preference for a smaller quantity and ignore their ‘natural’ preference for the larger amount of food [18,19,20].

While the aforementioned studies have provided increasing evidence for dogs’ human-like susceptibility to social influence, it still remains unclear whether their subsequent behaviour could be influenced by affiliative stimuli (eye-contact, petting) as primers to prosocial predispositions. Thus, in a series of experiments we tested dogs with different pre-treatments enacted by the owner to see if such social primers would influence subsequent behaviour in a social susceptibility task. To shed light on the mechanisms of canine social susceptibility, the same experimental paradigm was used to test the effect of intranasal oxytocin treatment. It has been suggested that positive social interactions with the owner increase endogenous oxytocin levels of both the dogs and their owners [21,22,23], although recent research suggest that such effects are conditional on the life experiences of the individuals [24]. At the same time, intranasal oxytocin administration has been shown to influence many aspects of dogs’ social behaviour (e.g., affiliative behaviour [25], positive expectation bias [26], use of human pointing [27], processing of emotional faces [28], gazing and social proximity [24]). Thus, we hypothesised that dogs’ social susceptibility would be influenced in the same way by both the owner’s social stimulation as well as intranasal oxytocin.

## 2. Experiment I

In the first experiment, using Prato-Previde et al.’s [18] ‘quantity preference’ paradigm, we investigated the hypothesis that a 10-min-long ‘socially stimulating’ pre-treatment, as compared to an ‘ignoring’ pre-treatment, eliminates dogs’ natural preference for the larger amount of food. Since the salient ostensive addressing signals accompanying human demonstration are likely to strongly affect the choice behaviour of dogs (c.f. [29]), the effect of different pre-treatments on dogs’ social susceptibility was tested in both communicative-addressing and non-communicative conditions (i.e., the experimenter either used ostensive addressing signals or avoided the use of these cues when expressing her preference for a smaller quantity).

### 2.1. Materials and Methods

#### 2.1.1. Ethical Statement

This research was approved by the National Animal Experimentation Ethics Committee (Ref No. XIV-I-001/531-4-2012). In accordance with ethics approval, all dog owners provided informed written consent to participate in the study and the research was done in accordance with the Hungarian regulations on animal experimentation and the guidelines for the use of animals in research described by the Association for the Study Animal Behaviour (ASAB).

#### 2.1.2. Subjects 

One hundred and fifteen task-naïve adult (>1 year; mean age ± SD: 4.1 ± 2.9 year, range: 1–16 years) pet dogs (57 males, 58 females; 96 purebreds from 41 different breeds and 19 mongrels; mean age ± SD: 3.81 ± 2.7 years) and their owners were recruited on a voluntary basis from the dog owner database of the Family Dog Project. Only those dogs were included in the test trials (*Human influence* trials) that met the criteria for Pre-training and showed a preference for the bigger amount of food in the *No Influence* phase of the quantity discrimination task (see below in Procedure). On the basis of this selection criterion, 33 dogs (28.7% of the total sample) were excluded from the experiment and the final sample consisted of 82 dogs (44 males, 38 females; mean age ± SD: 4.1 ± 2.96; 69 purebreds from 35 different breeds and 13 mongrels). These subjects were assigned to four experimental groups (see Section 2.1.3. Procedure) so that the distribution of age, sex and training experience did not differ by condition (see Appendix A).

#### 2.1.3. Procedure

The experiment took place in two different locations according to the dog owners’ availability: in a testing room at the Department of Ethology, Eötvös Loránd University, Budapest, Hungary (6 m × 3 m) and in a same-sized testing room in a dog training school. The behaviour of the dog and the owner was videotaped, and the choice behaviour of the subjects was analysed later. The owner was asked to refrain from feeding his/her dog at least 4 h prior to the test.

Familiarization. The owner (hereafter O) entered with his/her dog into the testing area and the dog was allowed to freely explore the environment for 1 min. Then, the experimenter (hereafter E) placed a few pellets of dry dog food on a yellow plastic plate (17 cm in diameter) and offered it to the dog by putting down the plate on the floor and encouraging the dog to take it. This phase served to familiarize the dogs with the experimental situation and to determine whether the dog could be motivated by food pellets. Different types of dry dog foods were used depending on the size of the dog (Happy Dog Adult Medium or Acana Adult Small Breed). If the dog ate the offered food pellets, it was followed by a pre-training phase. No dog was excluded in this phase.

Pre-training. The owner was asked to sit down on a chair at a predetermined point and to hold the dog there by its collar facing the middle of the room. Then, the E approached the dog with two identical yellow plastic plates (17 cm in diameter), one baited (with 4 pellets on it) and one empty, in her hands, stopped 1.5–2 m away from the dog (depending on its body size), and placed the plates simultaneously 1 m apart, equidistant from the dog. Then, she stepped back half a meter along the midline in between the plates and remained motionless while avoiding looking at the plates or into the dog’s eyes. At the moment at which the dog oriented itself towards the area between the two plates, the O was allowed to release and encourage the dog (“Go! It is yours!”), without using any other commands or gestures. Subjects were allowed to select only one plate. While the dog was eating the content of the chosen location, the non-chosen plate was removed by E. We repeated the pre-training trials with alternating the side of the food until the dog selected the baited plate twice in a row. One individual who did not meet this criterion after 8 trials was excluded from the study.

Pre-treatment (Figure 1a). Following the pre-training, subjects participated in either the socially stimulating (STIM) or the ignoring (IGN) pre-treatments. A validation of the endogenous oxytocin increasing effect of this exact socially stimulating pre-treatment has been previously published [30]. The 10-minute-long pre-treatment sessions took place in the same room as the pre-training trials.

*Socially stimulating pre-treatment* (STIM): O was asked to tether his/her dog with a 1.5-m-long leash to a plastic hook fixed on the floor. Then, O was seated within arm’s reach from the dog with E seated about 1 m away facing him/her. In the first half of the pre-treatment, O was asked to make eye contact with the dog as many times as possible and to slowly pet the dog in the meantime (5 min). In the second half of the pre-treatment, O was asked to initiate low-intensity play with his/her dog using a standard dog toy provided by E. O was encouraged to hide the toy behind his/her back and asked the dog to look at him/her. At the moment at which the dog established eye contact with O, the toy was given to the dog and it was praised and petted by O. Note that E avoided eye contact with the dog, and only talked to the O if she/he needed further instructions.

*Ignoring pre-treatment* (IGN): Arrangement of the participants was the same as in STIM; however, O was not allowed to interact with the dog. He/she was asked to fill in a questionnaire (not used for data collection purposes) about his/her dog’s social behaviour and behavioural habits without making eye contact, talking to or petting the dog. During this, the dog could play alone with the same dog toy used in STIM.

Quantity discrimination task. Pre-treatment was immediately followed by a task in which the dogs were presented with a choice between large (8 pellets) and small (1 pellet) food quantity. During this, we followed the procedure outlined in [18].

*No influence (NI) trials* (Figure 1b): O was asked to sit down on a chair at the predetermined point (same as in pre-training) and to hold the dog there by its collar facing the middle of the room. Then, E entered the room and approached the dog with two identical plates (baited with 1 and 8 food pellets, respectively) in her hands so that the dog’s muzzle was approximately 10 cm from the plates. This was done to ensure that the dog had the opportunity to inspect the content of both plates. The subsequent procedure was the same as in the pre-training. The *No Influence* phase consisted of six trials during which the left/right position of the plates containing large and small amount of food was counterbalanced. The purpose of the *No Influence* trials was twofold: first, to select those dogs that show a spontaneous preference for the larger food quantity and second, to gain baseline ‘choice behaviour’ data by which to measure change in response to social influence. Therefore, only those dogs were included in the Test trials that chose the larger amount of food (8 pellets) at least four times out of the six trials.

*Human influence (HI) trials* (Figure 1c): *No influence* trials were immediately followed by six additional *Human Influence* trials in which the setup and the procedure were basically the same as in the *No Influence,* except that the dog was allowed to choose only after observing the E expressing a preference for the small food quantity. That is, after having placed the plates containing small and large amount of food on the floor, E approached the plate containing only one piece of pet food, picked up the pellet and, with an enthusiastic tone of voice, said: “Hmm! Yummy!”. Then, she placed it back on the plate, stepped back half a meter along the midline in between the plates and remained motionless while avoided looking at the plates or into the dog’s eyes. Similarly to the *No Influence* trials, the position of the plates containing large and small food quantities was counterbalanced.

Subjects of both pre-treatment groups (STIM/IGN) participated in either addressing (ADDR) or non-addressing (NADDR) demonstration contexts. In the ADDR condition, the experimenter initiated eye contact with the dog when expressing her preference for the small amount of food and thus she clearly indicated that she is talking to the subject. In the NADDR condition, however, the experimenter avoided making eye contact with the dog (looked down) throughout the trial. 

### 2.2. Data Analysis

Two behaviour variables, a *food preference* score (*FPS*) and a *change-in-bias* score, were used for our analyses. 

*Food Preference* Score (*FPS*): Dogs received score 1 or 0 depending on whether they chose the plate containing the larger (1) or the smaller (0) quantity of food. A choice was noted when the dog touched one of the plates with its muzzle/nose. *FPS* was calculated by summing up the scores in the six *No Influence* trials (FPS-NI) as well as in the six *Human Influence* trials (FPS-HI).

*Change-in-bias* score: To assess dogs’ social susceptibility, that is, the effect of the experimenter’s preference for the smaller quantity on their behaviour, *change-in-bias* score was calculated by subtracting FPS-HI from FPS-NI score. Larger values for *change-in-bias* indicate a greater influential effect of the human demonstrator, that is, a decrease in the dogs’ preference for the larger amount of food.

To assess inter-observer agreement, a second person blind to the pre-treatment condition coded a randomly selected sample of 25% of the subjects. Cohen’s kappa value was 0.985 showing a high level of reliability for dogs’ choice behaviour.

A Generalized Linear Model analysis (ordinary logistic) was used to test the effects of pre-treatments (STIM vs. IGN) and demonstration contexts (ADDR vs. NADDR) on the dogs’ *change-in-bias* scores. Mann–Whitney tests with both *food preference* and *change-in-bias* scores were used for pair-wise group comparisons. Moreover, in order to determine whether dogs showed a tendency to switch towards a counterproductive behaviour in response to the human demonstration, we compared dogs’ *change-in-bias* scores in the four conditions to zero level (unchanged) using one-sample Wilcoxon signed-rank tests (SPSS 18.0 statistical package; IBM, Chicago, IL, USA).

### 2.3. Results and Discussion

The GLM analysis of the dogs’ *change-in-bias* showed a significant main effect of pre-treatment (STIM vs. IGN, χ^2^ = 8.861, *p* = 0.003) but no effect of demonstration context (ADDR vs. NADDR, χ^2^ = 0.704, *p* = 0.402). Importantly, however, we found a significant interaction between pre-treatment type × and demonstration context (χ^2^ = 3.918, *p* = 0.048) (Figure 2). In line with this, no difference was found in the ADDR demonstration context between STIM vs. IGN pre-treatments (Mann–Whitney test, U = 138.5, *p* = 0.446), but in the NADDR context, when ‘on-line’ ostensive addressing signals were not presented, subjects in the STIM group achieved a higher *change-in-bias* score compared to the IGN group (U = 120.5, *p* = 0.001).

Nevertheless, one-sample Wilcoxon signed rank tests indicated that dogs in all conditions changed their natural preferences for the larger quantity and conformed to the human’s counterproductive choice (i.e., the *change-in-bias* scores were significantly above zero in each group: STIM-ADDR, *Z* = 3.671, *p* < 0.0001; STIM-NADDR, Z = 4.046, *p <* 0.0001; IGN-ADDR, Z = 3.072, *p* = 0.002; IGN-NADDR, Z = 2.113, *p* = 0.035).

Further analysis of the *FPS* showed no difference between pre-treatment groups (STIM vs. IGN) in how many times they chose the larger food quantity in the *No Influence* trials (ADDR demonstration context: U = 155.5 *p* = 0.84; NADDR demonstration context: U = 218, *p* = 0.287). In the subsequent *Human Influence* trials, however, demonstration context affected dogs’ performance. That is, in the NADDR demonstration context dogs after STIM pre-treatment chose significantly less often the larger food quantity than dogs who received IGN pre-treatment (U = 154.5, *p* = 0.012), but when human demonstration was ostensively addressed to them (ADDR context), there was no difference between STIM and IGN pre-treatment groups (U = 136.5, *p* = 0.42).

The results of this experiment corroborate the notion that the food choice behaviour of dogs can be highly influenced by the expression of a human’s preference even when the demonstrator is unfamiliar to the dog (see also [19]), and even if dogs are misled by human towards a less favourable choice. The question of whether or not dogs are ostensively addressed during demonstration has been attributed a minor importance [20], and our results also seem to support this conclusion, as subjects in both addressing and non-addressing demonstration contexts showed positive (i.e., significantly above zero) *change-in-bias* scores.

However, in addition to the potential role of ostensive addressing during demonstration, in the present experiment, we paid particular attention to an additional contextual factor that could influence dogs’ social susceptibility in such situations: the impact of social priming (visual and tactile stimulation) on dogs’ behavioural susceptibility. The fact that subjects were more likely to follow the experimenter’s preference in the test phase after tactile and visual stimulation by the owner provides support for the effect of social priming as a social influencing mechanism. It is also important to note that although the addressing vs. non-addressing demonstration context had no main effect on the dogs’ tendency to conform to the human’s choice, the significant *pre-treatment* × *demonstration context* interaction suggests that the salient ostensive addressing signals accompanying human demonstration could reduce or even eliminate social priming effects in dogs.

In summary, these results suggest that social-affiliative stimuli provided by the owner has the potential to increase dogs’ tendency to conform to an unfamiliar experimenter’s food preference in a quantity discrimination task. This finding indicates that the social priming effect—the effect of social interaction with the caregiver on subsequent decision-making processes—may also occur in dogs. This is in line with human studies indicating that affiliative stimuli can act as primers for prosocial predispositions in humans [31,32].

## 3. Experiment II

Recent evidence suggests that oxytocin has the potential to make the brain more receptive to social information; this neuropeptide modulates cooperation (see [33] for a review) and may stimulate in-group conformity in humans [34]. Positive social interaction is sufficient to increase central oxytocin level in human subjects (e.g., [35]), and it has also been shown that social-affiliative stimuli (soft talk, long smooth strokes, grooming and low-key playing with a human) enhance dogs’ peripheral oxytocin level [22,36,37]. The role oxytocin plays in dogs’ social–cognitive processes is further supported by the findings that polymorphisms in the OXTR gene are related to human directed social behaviour in dogs [38] and intranasal oxytocin administration can induce higher social orientation and affiliation [25], as well as increase positive expectations in dogs in a social learning task [26].

These studies raise the possibility that there is a link between changes in central oxytocin level and social susceptibility and/or social synchronisation mechanisms in the dog. Thus, the purpose of Experiment II was to investigate whether intranasal administration of exogenous oxytocin as compared to placebo treatment could increase dogs’ tendency to conform to the human’s counterproductive choice in the quantity discrimination task.

### 3.1. Materials and Methods

On average, 6–7 months after the first experiment, dogs were again observed in the quantity discrimination task. In this case, however, subjects received synthetic oxytocin or placebo treatment before the *Human Influence* trials. As the first experiment indicated that salient addressing signals from the human demonstrator may counteract the effects of socially stimulating pre-treatment, dogs in this second study were presented with non-addressing demonstration context (without making eye contact and gaze-shift when making a counterproductive choice). We investigated whether exogenous oxytocin administration caused a bias in dogs’ choice behaviour similar to that found in the socially stimulating pre-treatment (STIM) group of the Experiment I.

#### 3.1.1. Subjects

Only 37 adult pet dogs (20 males, 17 females; 28 purebreds from 17 different breeds and 9 mongrels; mean age ± SD: 4.45 ± 2.8 years, range: 1–16 years) and their owners from the 82 subjects that participated in Experiment I could be persuaded to participate in this study. The dogs were assigned in a double-blind manner to receive either intranasal oxytocin (OT) or placebo (PL) treatment so that both the experimenter and the owner were unaware of the type of treatment. Subjects from the STIM and IGN pre-treatment groups (see Experiment I) were assigned evenly to the OT and PL groups (Fisher’s exact test, *p* = 0.73). Like in Experiment I, only those dogs were included in the *Human Influence* trials that showed a preference for the bigger amount of food in the *No Influence* phase of the quantity discrimination task. On the basis of this selection criterion, 3 dogs were excluded from the experiment and two additional dogs also had to be excluded because they showed fear/avoidance during intranasal administration of OT/PL spray. The final sample consisted of 32 dogs (18 males, 14 females; mean age ± SD: 4.56 ± 2.92 years; 25 purebreds from 14 different breeds and 7 mongrels). Subjects in the OT and PL group did not differ in age, sex ratio, proportion of trained vs. un-trained individuals, time elapsed between Experiment I and II, nor in their choice behaviour (*change-in-bias* score) in Experiment I (see Appendix A).

#### 3.1.2. Procedure

The experiment took place in the same testing room, and the experimental procedure had basically the same structure as in Experiment I, in this case, however, another female experimenter who was unfamiliar to the dog participated. *Familiarization* and *Pre-training* (see procedure section above in Experiment I) were followed by the *No influence* trials of the quantity discrimination task. After the six *No Influence* trials, those dogs who showed a preference for the larger amount of food (i.e., chose the 8 pellets at least four times), received a single intranasal dose of 12 IU oxytocin (Syntocinon-Spray, Novartis; nasal spray with a nebulizer) or placebo (isotonic sodium chloride 0.9% solution). The intranasal administration (1 and 2 puffs per nostril) was conducted by an assistant to prevent the association between discomfort/anxiety and the experimenter. Then, a 45-minute waiting period followed, which was presumed to be necessary for the central oxytocin levels to reach a plateau [39]. During this time, dogs went for an on-leash walk with their owners and the experimenter, who ensured that the owner did not make any social contact with the dog (e.g., did not make eye contact, did not talk to it, did not pet it) and kept the length as well as the speed of the walk as standard as possible. This OT pre-treatment procedure has been validated for dogs by showing that oxytocin as compared to placebo decreases heart rate and increases heart rate variability [40]. Additionally, in a sample of beagle dogs, it has been confirmed that both serum and urinary oxytocin levels increase following nasal oxytocin administration [41]. Pre-treatment was followed by the *Human Influence* trials (6 trials) of the quantity discrimination task where the E expressed a preference for the small food quantity in a non-addressing manner (i.e., avoiding eye-contact with the dog—for the procedure see the NADDR demonstration context in Experiment I).

#### 3.1.3. Data Collection and Analysis

Like in Experiment I, *FPS* (in both *No Influence* and *Human Influence* trials) were video recorded, and these records were used to calculate *change-in-bias* scores for each dog. Mann–Whitney tests were used to compare the effects of OT vs. PL treatments on the dogs’ *food preference* (*FPS*) and *change-in-bias* scores (SPSS 18.0 statistical package; IBM, Chicago, IL, USA). Furthermore, to investigate potential within-subject associations between dogs’ choice behaviour in Experiments I and II, partial correlation analyses (controlling for the change of demonstration context and pre-treatment) were conducted on *FPS* and *change-in-bias* scores of dogs in Experiments I and II.

### 3.2. Results and Discussion

Dogs assigned to the oxytocin treatment group showed a non-significant trend towards higher *FPS* in the *No Influence* trials (OT vs. PL groups, Mann–Whitney test, U = 81.5, *p* = 0.082), and there were also no significant differences in the *FPS* during the *Human Influence* trials (OT vs. PL groups, Mann–Whitney test, U = 98.5, *p* = 0.278). However, the analysis of *change-in-bias* scores indicates significant differences between OT and PL treatment groups. Namely, dogs, after having received intranasal administration of oxytocin showed a higher tendency to give up their preference for the larger quantity, and thus they were more willing to conform to the human’s choice (OT vs. PL, U = 72, *p* = 0.036) (Figure 3).

No significant correlations were found between subjects’ choice behaviour in Experiment I and Experiment II with respect to their *food preference* scores (*No Influence* trials: r = 0.269, *p* = 0.889; *Human Influence* trials: r = 0.369, *p* = 0.058) and *change-in-bias* scores (r = 0.210, *p* = 0.293).

The finding that dogs, after administration of oxytocin, showed an increased responsiveness to the *Human Influence* suggests that social priming and exogenous oxytocin may have a similar effect on dogs’ social susceptibility and raises the possibility of shared neurocognitive mechanisms across these different pre-treatment (i.e., STIM and OT) conditions. This conclusion fits with the findings of human–dog interaction studies reporting a fundamental role of oxytocin in affiliative contexts [22,36,42]. Furthermore, the fact that subjects’ performances in Experiments I and II were unrelated suggests that the effect of pre-treatment was greater than that of individual variation in dogs’ social susceptibility.

## 4. General Discussion

Previous research has shown that positive social interactions with (even unfamiliar) human partners have the potential to impact physiological responses in dogs (stress relieving effect [43]). More recently, it has also been demonstrated that the exposure to positively—versus negatively—valenced interactions with an unfamiliar human has differential effects on dogs’ subsequent behaviour in a situation where an unfamiliar human requests an out-of-reach object [44]. These studies raise the interesting possibility that ‘social priming effect’ can affect dogs’ behaviour. In line with these findings, here we aimed to investigate how priming with social stimuli (i.e., social stimulation provided by the owner) and pre-treatment with the neurohormone oxytocin affect dogs’ propensity to tune in to an unfamiliar human demonstrator’s counterproductive choice. Although previous studies have shown that dogs readily adopt counterproductive responses in object- or food-choice tasks as a result of observations of human demonstrations [18,45,46], the social-communicative signals in those experiments were presented as an integral part of the demonstration context. Our study provides the first direct experimental evidence that dogs can also be primed with social-communicative stimuli provided by the owner, and the social priming effect also works across contexts/situations (i.e., when the pre-treatment procedure is unrelated to the subsequent task situation).

The results of Experiment I are in line with the observations that priming with affiliative stimuli can enhance prosocial predispositions in humans [31,32], and thus have important implications for further research on dog–human interactions. Furthermore, our study highlights a potentially important, yet largely neglected methodological issue in the study of dogs’ social cognition. Namely, most of the research has to date paid little attention to the potential effects of social stimuli that the subjects receive during the preliminary (warm-up/familiarization) phase of an experiment. Our results clearly show that such stimulation can have a strong influence on dogs’ social-cognitive functioning in the test phase, and thus social interactions prior to any study procedure need to be controlled by the experimenters.

The results also draw attention to the role of ostensive addressing signals (eye contact) accompanying the demonstration which can eliminate the effect of social priming. This is in agreement with previous findings (see [47] for a review) showing that the social-communicative nature of the task has a great influence on the dog’s task performance. There is increasing evidence to suggest that eye contact and verbal addressing have a key role in inducing specific responsiveness in dogs [48,49,50]. For dogs human ostensive signals indicate that they are being instructed and they can be strongly affected by such signals even in those situations in which these signals serve to highlight an inefficient or mistaken solution [29,45,51].

The second experiment provides support for a probable role of oxytocin in the neurocognitive mechanism mediating the social priming in dogs. This is in line with recent evidence from human research showing that oxytocin increases trust [52], cooperation [53], and affiliative attitudes [54] towards others. There is, however, no information yet about the underlying neural mechanisms and hormonal interactions that are triggered by exogenous/endogenous oxytocin increase and thus serving as a direct mediator of the observed behavioural changes.

It is worth mentioning, however, that despite the widespread use of oxytocin nasal spray in social interaction studies, the exact mechanism and pathway through which exogenous oxytocin reaches its receptors in the brain is not fully understood [55]. Although Neumann et al. [56] showed that after intranasal administration of oxytocin both peripheral and central oxytocin levels increased in rodents, the validity of this method is based mainly on circumstantial evidence. For example, intranasal oxytocin has been shown to increase heart rate variability in both humans [57] and dogs [40]. Peripheral (salivary) vasopressin levels in humans elevate 15 min. after intranasal administration and reach a plateau between 45–120 min. [58]. In beagle dogs, serum oxytocin peaked 15 min, while urinary oxytocin peaked between 45 and 60 min after intranasal administration [41].

Moreover, although a previous study [37] suggested that both eating and exercise increase dogs’ oxytocin secretion, in our study, dogs in the different pre-treatment conditions participated in the same amount of physical activity and there was no difference in their food intake during the *No Influence* phase (as there was no difference in their choice behaviour). Thus, in contrast to the obvious methodological uncertainties in using exogenous oxytocin, we suggest that this method has the potential to gain a better insight into dogs’ functionally human-analogue social cognition. We should note, however, that it is still unclear whether the aforementioned pre-treatment effects and the potential role of oxytocin in social susceptibility is a dog-specific phenomenon or other domesticated (e.g., cats) and/or non-domesticated social animals (e.g., wolves) are also susceptible to such priming effects.

Concerning the functional role of oxytocin in the social priming analogous results were obtained by Kis et al. [59] in humans. They found that intranasally administered oxytocin and pre-exposure to social stimuli (physical and eye contact with a female) have similar effects on the perception of negative facial emotions in adult male participants. That is, these participants rated negative emotional faces more positively than non-socially pre-treated or placebo groups. These findings are consistent with the notion that positive social stimuli can have an oxytocin-releasing effect in the central nervous system [35] and then a feed-forward mechanism (mediated by autoreceptors on oxytocin neurons [60]) facilitates social-affiliative behaviours toward others.

## 5. Conclusions

In summary, dogs’ social susceptibility in this ‘counterproductive influence’ task seems analogous to a complex human behaviour that has been shown to be influenced by oxytocin—social susceptibility [34]. However, we should be cautious when drawing a parallel between dogs and humans with respect to social susceptibility. First, it is unclear whether or not the observed functional analogy between the dogs’ social suggestibility and the human tendency to susceptibility based on similar social-cognitive mechanisms, and second, our results provide no direct evidence that social priming or intranasal oxytocin administration enhanced the dogs’ central oxytocin level thereby influencing the dogs’ social receptivity.

## Figures and Tables

**Figure 1 animals-12-00296-f001:**
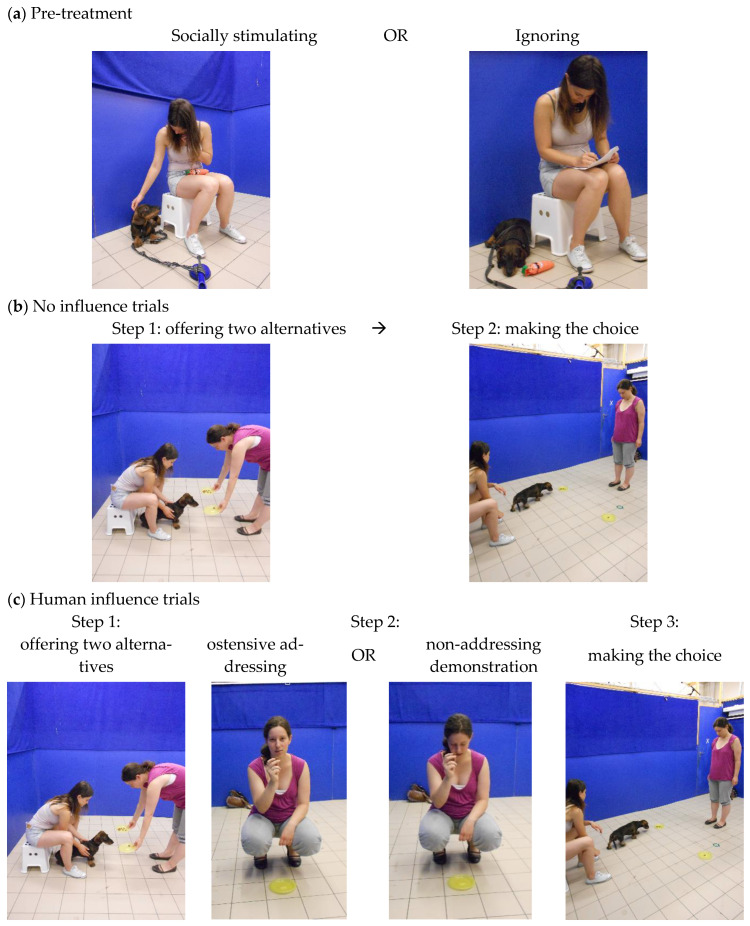
Photo illustration of the main phases of the experimental procedure. (**a**) Different types of pre-treatments. (**b**) Different steps of the ’No influence’ trials in the quantity discrimination task (trials 1–6). (**c**) Different steps of the ‘Human influence’ trials in the quantity discrimination task (trials 7–12).

**Figure 2 animals-12-00296-f002:**
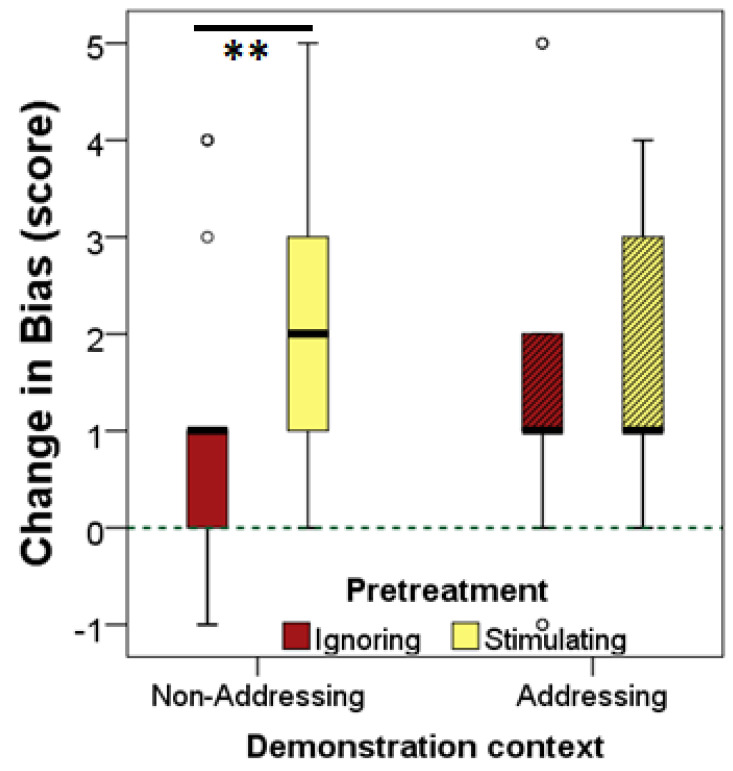
*Change-in-bias* scores of the four experimental groups in Experiment I. Larger values indicate a greater influential effect of the human demonstrator (preference for the smaller amount). Medians are represented by bold lines; boxes indicate the lower and upper interquartile range and whiskers extend to the smallest and largest values excluding outliers and extremities. Asterisks (**) indicate significant difference between dogs received socially stimulating (STIM) and ignoring (IGN) pre-treatment in the non-addressing demonstration context at *p* < 0.01.

**Figure 3 animals-12-00296-f003:**
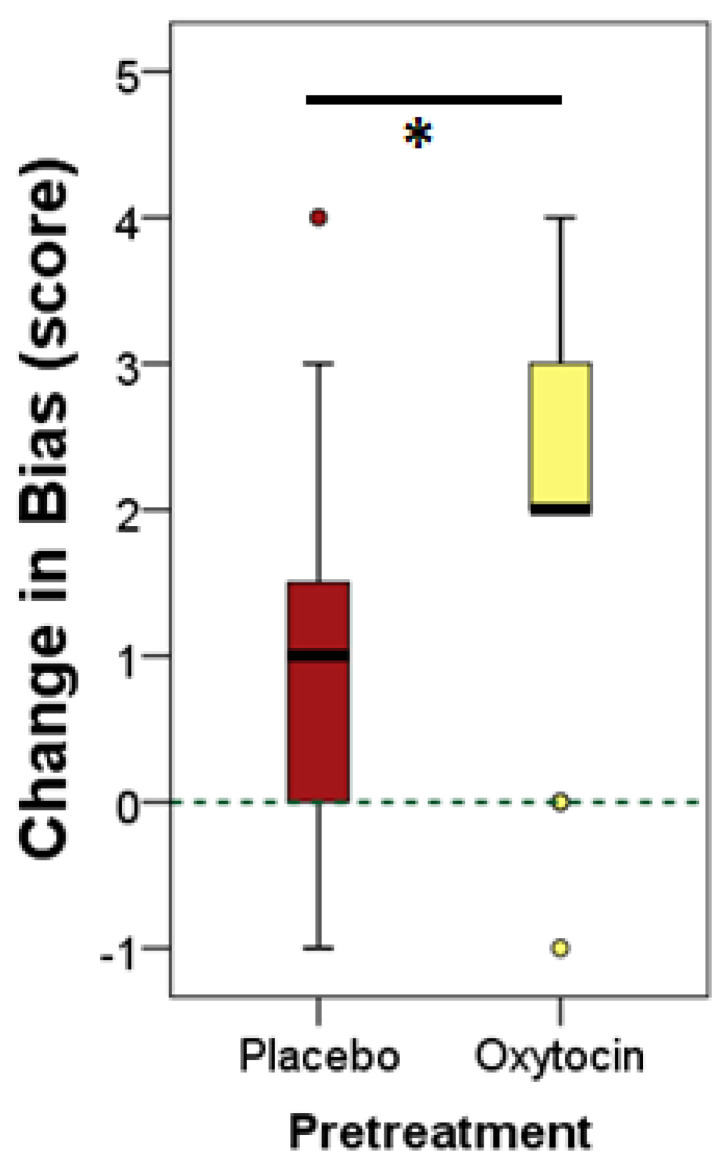
*Change-in-bias* scores of the oxytocin and placebo (isotonic sodium chloride) pre-treatment groups in Experiment II. Medians are represented by bold lines; boxes indicate the lower and upper interquartile range and whiskers extend to the smallest and largest values excluding outliers and extremities. Asterisk (*) indicates significant difference between dogs received intranasal oxytocin and placebo pre-treatment at *p* < 0.05.

## Data Availability

All data are stored safely in the institutional database (NAS server) of the Research Centre for Natural Sciences and is available upon request from the authors.

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
