# Peer review of "Social Stimulation by the Owner Increases Dogs’ (*Canis familiaris*) Social Susceptibility in a Food Choice Task—The Possible Effect of Endogenous Oxytocin Release"

_animals, 2022, doi:10.3390/ani12030296_

Round 1

Reviewer 1 Report

This study was conducted on the hypothesis that social stimulation from owners would increase oxytocin in dogs, thus making the dogs more susceptible to human behavior. As the results, both social stimulation from the owner and oxytocin administration caused the dogs to make choices that were influenced by the experimenter's behavior. As hypothesized, canine social cognitive behavior may change depending on how they interact with humans.

The results are clear, and the logic is easy to understand. I found the findings very useful for future social cognition experiments in animals. However, I was concerned about the following two points.

First, I was concerned about the definition of the word "conformity" used in the title and in the text. I think "conformity" as the technical term is defined as copying a behavioral decision made by the majority (van Leeuwen and Haun, 2014, https://doi.org/10.1016/j.anbehav.2014.08.004). However, in this paper, the choice was not made by the dogs based on a specific method presented by multiple people. It would be better to use other words (social influence, social susceptibility, etc.) to avoid misunderstandings.

Secondly, in recent years, oxytocin administration has been increasingly used in social cognitive studies. However, because of the various effects of oxytocin, there is a danger that oxytocin can actually be explained by somehow fitting it into the interpretation of social and cognitive behavior. For example, can the choice behavior of the dogs in this experiment be explained by the direct effect of oxytocin? The behavior and cognition of animals, including humans, can vary greatly depending on the characteristics of the individual, social context, and other endocrine states. In experiment 1, the dogs were stressed by being ignored by their owners, i.e., the effect of cortisol may have reduced the social influence from humans, and in experiment 2, the administration of oxytocin lowered cortisol concentration, so the dogs may have been influenced by humans. This means that oxytocin is not directly involved in the results of Experiment 1. Since physiological indices were not collected in this experiment, I understand that this cannot be confirmed this time. Maybe in your future study you could add other factors to your analysis.

The following are minor points.

L.102

Please also provide the age range of dogs. In Table S1, I was concerned that, although there is no statistically significant difference, the mean value is slightly higher in the stimulating group.

L.154

Why was the experimenter present with the owner and the dog?

Was the experimenter also present at the IGN-treatment? The difference of social context may influence oxytocin levels. In reference 30, it appears that the experiment was conducted with just the canine and the owner.

L.233

Please insert (Fig. 2) in the relevant part of the results.

There are many abbreviated words which makes  it confusing. Please consider how to abbreviate them and whether there are cases where you don't need to.

For example, STIM and ADDR are not immediately understandable at first glance.

I am interested in how long the effects of pre-treatment and oxytocin last (cf. Simona et al., 2021, https://doi.org/10.1016/j.applanim.2021.105534). How did your results for the 6 trials of each test change? Did any of the dogs choose the large food (8 pellets) halfway through?

Author Response

Dear Editor,

Thank you for your assessment of our manuscript animals-1531668 entitled “Social stimulation by the owner increases dogs’ (Canis familiaris) social conformity in a food choice task. The possible effect of endogenous oxytocin release”.

Please find below our detailed answers to the reviewers’ comments which helped us improve the manuscript for this revision.

Sincerely,

Anna Kis

(on behalf of all authors)

Reviewer 1

This study was conducted on the hypothesis that social stimulation from owners would increase oxytocin in dogs, thus making the dogs more susceptible to human behavior. As the results, both social stimulation from the owner and oxytocin administration caused the dogs to make choices that were influenced by the experimenter's behavior. As hypothesized, canine social cognitive behavior may change depending on how they interact with humans.

The results are clear, and the logic is easy to understand. I found the findings very useful for future social cognition experiments in animals. However, I was concerned about the following two points.

Thank you for this positive assessment and the helpful suggestions. Please see below our response to both main points as well as the minor issues.

First, I was concerned about the definition of the word "conformity" used in the title and in the text. I think "conformity" as the technical term is defined as copying a behavioral decision made by the majority (van Leeuwen and Haun, 2014, https://doi.org/10.1016/j.anbehav.2014.08.004). However, in this paper, the choice was not made by the dogs based on a specific method presented by multiple people. It would be better to use other words (social influence, social susceptibility, etc.) to avoid misunderstandings.

We concur the reviewer’s suggestion and change the term to “social susceptibility” throughout the manuscript.

Secondly, in recent years, oxytocin administration has been increasingly used in social cognitive studies. However, because of the various effects of oxytocin, there is a danger that oxytocin can actually be explained by somehow fitting it into the interpretation of social and cognitive behavior. For example, can the choice behavior of the dogs in this experiment be explained by the direct effect of oxytocin? The behavior and cognition of animals, including humans, can vary greatly depending on the characteristics of the individual, social context, and other endocrine states. In experiment 1, the dogs were stressed by being ignored by their owners, i.e., the effect of cortisol may have reduced the social influence from humans, and in experiment 2, the administration of oxytocin lowered cortisol concentration, so the dogs may have been influenced by humans. This means that oxytocin is not directly involved in the results of Experiment 1. Since physiological indices were not collected in this experiment, I understand that this cannot be confirmed this time. Maybe in your future study you could add other factors to your analysis.

We agree with the reviewer that the conclusion of the current study is limited to say that oxytocin versus placebo as well as social stimulation versus ignoring caused a certain change in dogs’ behaviour, but we do not have any information about the underlying intermediate mechanisms. We have briefly included this limitation to the discussion (lines 447-450).

The following are minor points.

L.102

Please also provide the age range of dogs. In Table S1, I was concerned that, although there is no statistically significant difference, the mean value is slightly higher in the stimulating group.

We have now included the age range for Experiments I & II also in the main text. Regarding the detailed data in the Supplementary Material, as the reviewer notes, there is no statistical difference between groups and the mean±SE ranges of the different treatment groups overlap in all cases.

L.154

Why was the experimenter present with the owner and the dog?

Was the experimenter also present at the IGN-treatment? The difference of social context may influence oxytocin levels. In reference 30, it appears that the experiment was conducted with just the canine and the owner.

The experimenter was present to ensure that the pre-treatment protocol was followed properly by the owner. The arrangement of the participants was the same in both the STIM and IGN pre-treatment (with O and E sitting in the same positions) as stated in line 163.

L.233

Please insert (Fig. 2) in the relevant part of the results.

Figure 2 is referenced in line 239.

There are many abbreviated words which makes it confusing. Please consider how to abbreviate them and whether there are cases where you don't need to. For example, STIM and ADDR are not immediately understandable at first glance.

Please note that we have changed some of the abbreviations in line with the detailed suggestions of Reviewer 2. However, we would prefer not to reduce the number of abbreviations as the long names / definitions of task conditions and variables would make the text hard to read.

I am interested in how long the effects of pre-treatment and oxytocin last (cf. Simona et al., 2021, https://doi.org/10.1016/j.applanim.2021.105534). How did your results for the 6 trials of each test change? Did any of the dogs choose the large food (8 pellets) halfway through?

The change-in-bias scores we observed varied greatly among individual dogs: from -1 (choosing the bigger amount one time more than in baseline) to +5 (choosing the smaller amount five times more than in baseline). However, we did not see any systematic effect of trial number on dog’s choice behaviour.

Reviewer 2 Report

The study is focused on the effect of social stimulation (experiment I) and oxytocin administration (experiment II) on a dog's social conformity during a food choice task.

The topic of the study is very interesting, and the results obtained are helpful for the comprehension of the dog-human relationship. Knowing better the possible behavioral effect of the social stimulation by the owner could be interesting, especially in the training context where humans and dogs work together. Focus on the possible role of intranasal oxytocin is useful to better understand the role of this neurohormone in the behavioral performance of dogs (and probably of other domesticated and undomesticated species).

The paper is well written and conceptually clear. The references are appropriate, and the results are well discussed.

It might be useful to know the type of training (for example sport, defense, rescue) of the dogs indicated in Tables S1 and S2.

Some revisions are listed below:

Line 11: Add the comma after “Furthermore”.

Line 14: Add the comma after “study”.

Line 18: Replace “result” with “results”.

Line 22: Add the comma after “However”.

Line 24: Add the comma after “Experiment I”.

Line 25: Add the point after “et al”.

Lines 25-26: Remove “In Experiment I”.

Line 32: Add the comma after “Experiment II”.

Line 67: Add the comma after “Thus”.

Line 68: Replace “pretreatments” with “pre-treatments”.

Line 70: Add the comma after “conformity”.

Lines 75-77: Use the parenthesis from “e.g.” to “[24].”

Line 77: Add the comma after “Thus”.

Line 108: Add the comma after “criterion”.

Line 123: Add the comma after “Then”.

Line 129: Add the comma after “pellets”.

Line 133: Remove the extra space between “E” and “approached”.

Line 139: Remove the extra space between “O” and “was allowed”.

Line 147: Add the comma after “pre-training”.

Line 159: Add the comma after “with O”.

Line 166: Add the comma after “this”.

Line 221: Add the comma after “agreement”.

Lines 237 and 239: Add the spaces before and after “=”.

Line 238: Add the comma after “this”.

Line 312: Add the comma after “experiment”.

Line 330: Add the comma after “Experiment I”.

Line 332: Add the comma after “criterion”.

Line 353: Add the comma after “time”.

Line 369: Replace “food preference” with “FPS” and “change-in-bias” with “change-in-bias”.

Line 378: Replace “food preference” with “FPS”.

Line 380: Replace “change-in-bias” with “change-in-bias”.

Lines 394-395: Add the spaces before and after “=”.

Line 442: Use a single square bracket for all reference numbers.

Line 475: The closing parenthesis is missing.

Lines 39, 547, 549, 551, 615, 623, 630: Use the italics for the name of the species.

Author Response

Reviewer 2

The study is focused on the effect of social stimulation (experiment I) and oxytocin administration (experiment II) on a dog's social susceptibility during a food choice task.

The topic of the study is very interesting, and the results obtained are helpful for the comprehension of the dog-human relationship. Knowing better the possible behavioral effect of the social stimulation by the owner could be interesting, especially in the training context where humans and dogs work together. Focus on the possible role of intranasal oxytocin is useful to better understand the role of this neurohormone in the behavioral performance of dogs (and probably of other domesticated and undomesticated species).

The paper is well written and conceptually clear. The references are appropriate, and the results are well discussed.

We thank the reviewer for this positive assessment.

It might be useful to know the type of training (for example sport, defense, rescue) of the dogs indicated in Tables S1 and S2.

This information is unfortunately not available to us, participating owners only reported if their dog has undergone any training or not, but we did not ask for further details. We agree that there could be interesting differences based on the type of training received, however this study was not designed to investigate training effects and thus the current sample size would probably not be sufficient to neither to confirm nor to discard any hypotheses concerning this issue.

Some revisions are listed below:

Line 11: Add the comma after “Furthermore”.

Line 14: Add the comma after “study”.

Line 18: Replace “result” with “results”.

Line 22: Add the comma after “However”.

Line 24: Add the comma after “Experiment I”.

Line 25: Add the point after “et al”.

Lines 25-26: Remove “In Experiment I”.

Line 32: Add the comma after “Experiment II”.

Line 67: Add the comma after “Thus”.

Line 68: Replace “pretreatments” with “pre-treatments”.

Line 70: Add the comma after “conformity”.

Lines 75-77: Use the parenthesis from “e.g.” to “[24].”

Line 77: Add the comma after “Thus”.

Line 108: Add the comma after “criterion”.

Line 123: Add the comma after “Then”.

Line 129: Add the comma after “pellets”.

Line 133: Remove the extra space between “E” and “approached”.

Line 139: Remove the extra space between “O” and “was allowed”.

Line 147: Add the comma after “pre-training”.

Line 159: Add the comma after “with O”.

Line 166: Add the comma after “this”.

Line 221: Add the comma after “agreement”.

Lines 237 and 239: Add the spaces before and after “=”.

Line 238: Add the comma after “this”.

Line 312: Add the comma after “experiment”.

Line 330: Add the comma after “Experiment I”.

Line 332: Add the comma after “criterion”.

Line 353: Add the comma after “time”.

Line 369: Replace “food preference” with “FPS” and “change-in-bias” with “change-in-bias”.

Line 378: Replace “food preference” with “FPS”.

Line 380: Replace “change-in-bias” with “change-in-bias”.

Lines 394-395: Add the spaces before and after “=”.

Line 442: Use a single square bracket for all reference numbers.

Line 475: The closing parenthesis is missing.

Lines 39, 547, 549, 551, 615, 623, 630: Use the italics for the name of the species.

Thank you for this very detailed list of corrections, all of them have been implemented in the manuscript text.